# Efficacy and safety of palbociclib and ribociclib in patients with estrogen and/or progesterone receptor positive, HER2 receptor negative metastatic breast cancer in routine clinical practice

Sushmita Rath[1][ ⵟ], Prahalad Elamarthi[1][ ⵟ], Pallavi Parab[1], Seema Gulia[1], Ravindra Nandhana[1], Smruti Mokal[2], Yogesh Kembhavi[1], Prema Perumal[1], Jyoti Bajpai[1], Jaya Ghosh[1], Sudeep Gupta[1] *

1 Department of Medical Oncology, Tata Memorial Centre, Homi Bhabha National Institute, Mumbai, Maharashtra, India, 2 Department of Biostatistics, Tata Memorial Centre, Homi Bhabha National Institute, Mumbai, Maharashtra, India

ⵟ These authors contributed equally to this work.

* sudeepgupta04@yahoo.com

**Data Availability Statement:** There are federal Indian regulations on providing access to data for

## Abstract

### Background

There is scant data from India on efficacy and safety of palbociclib and ribociclib in routine clinical practice.

### Methods

This retrospective, observational, single institution study included patients with estrogen and/or progesterone receptor positive and human epidermal growth factor receptor 2 (HER2) negative metastatic breast cancers, who received palbociclib or ribociclib with any partner endocrine therapy in any line of treatment between January 2016 and June 2019. Data were analyzed for progression-free survival (PFS), overall survival (OS) and toxicity.

### Results

The study included 101 female patients with median age of 57 (IQR 48–62) years, of whom 80 (79.2%) were postmenopausal, 79 (78.2%) received palbociclib or ribociclib in second- or later-line treatment, 59 (58.4%) received fulvestrant and 41 (40.6%) received an aromatase inhibitor. In first-line treatment, at a median follow-up of 21.7 (0.5–41.9) months, median PFS and OS were 21.1 (95%CI 16.36-not estimable) months and not reached, respectively. In second- or later-line setting, at a median follow-up of 17.2 (0.5–43.7) months, median PFS and OS were 5.98 (95%CI 4.96–7.89) months and 20.2 (95%CI 14.1-not estimable) months, respectively. Grade 3–4 neutropenia and febrile neutropenia were seen in 45 (45.0%) and 9 (9.0%) patients, respectively while dose reduction was required in 32 (31.7%) patients. In multivariable Cox regression analysis, first-line setting (HR 0.49,

individuals outside the country. All such requests have to be cleared by the Indian Health Ministry Screening Committee. The de-identified data set will be provided to any interested researcher outside India after obtaining the required approval from the Health Ministry Screening Committee who can be contacted at hmscihdicmr@gmail. com. For researchers inside India the de-identified data can be provided with Institutional Ethics Committee approval. The Institutional Ethics Committee can be contacted at tmhethics@gmail. com.

**Funding:** The authors received no specific funding for this work.

**Competing interests:** Sudeep Gupta is a member of the advisory board at the following institutions: Roche, Sanofi, Dr. Reddy's Laboratories, Biocon, Pfizer, Oncosten, Core Diagnostics, and Astrazeneca. Jyoti Bajpai is a member of the advisory board at Novartis. Seema Gulia is a member of the advisory board at Novartis and Eisai. Sudeep Gupta has received research funding from Roche, Sanofi, Johnson & Johnson, Amgen, Celltrion, Oncosten, Novartis, Intas, Eisai, Biocon, and Astrazeneca. Sushmita Rath has received research funding from Astrazeneca and JSS clinical research. Seema Gulia has received research funding from Eli Lilly, Pfizer Inc, Celltrion, Kendle India Pvt Ltd, and Zydus. Jyoti Bajpai has received research funding from Eli Lilly, Novartis, Roche, Samsung Bioepis, Paxman, and Sun Pharma. The authors received no specific funding for this work. This does not alter our adherence to PLOS ONE policies on sharing data and materials. The authors would like to declare the following products associated with the theme of this research: palbociclib and ribociclib.

95%CI 0.25–0.97, p = 0.043) and ECOG performance status 1 (HR 0.43, 95%CI 0.20–0.91, p = 0.028) were significantly associated with PFS while only ECOG PS 1 was significantly associated (HR 0.04, 95%CI 0.008–0.206, p = 0.000) with OS.

## Conclusion

Palbociclib and ribociclib, when used in routine clinical practice in first or subsequent lines of treatment, resulted in efficacy and toxicity outcomes in concordance with those expected from pivotal trials.

## Introduction

The treatment of patients with estrogen receptor (ER) and/or progesterone receptor (PR) positive and human epidermal growth factor receptor 2 (HER2) negative, metastatic breast cancer (MBC) has undergone an evolution in the past few years with the introduction of cyclin dependent kinase 4/6 (CDK 4/6) inhibitors palbociclib, ribociclib and abemaciclib. These drugs are currently considered standard treatment in the first line and second-line settings in patients with MBC, in combination with an aromatase inhibitor (AI) and fulvestrant, respectively [1–8].

It is well known that, when used in routine clinical practice, outcomes and toxicity profile of new treatments may vary from those seen in pivotal trials due to patient selection, concomitant medications, pharmacogenomics and other factors [9, 10]. Therefore, appraisal of new treatments in routine practice is an important component of the evidence base. Although a few studies have analyzed the real-world outcomes with CDK 4/6 inhibitors [11–14] such data from India is scant [15, 16].

We performed a retrospective analysis to evaluate the efficacy and toxicity of the two available CDK 4/6 inhibitors in India, palbociclib and ribociclib, in patients with metastatic breast cancer treated at a single Centre in India.

## Methods

The study was conducted as a retrospective analysis of a single-Centre database after obtaining approval of the Institutional Ethics Committee. Waiver of consent was obtained from the Ethics committee because this was a retrospective analysis and the data was analyzed anonymously. Patients with histopathologically proven ER and/or PR positive and HER2 negative breast cancer who had clinical and/or radiological evidence of distant metastases or locally advanced/recurrent breast cancer, not amenable to curative intent local therapy, were included in the study. All patients were treated with palbociclib or ribociclib in combination with any other endocrine therapy partner and could be receiving this therapy in any line of treatment. Patients could have received any number of lines of prior systemic therapies before starting CDK 4/6 inhibitor. Palbociclib and ribociclib were started in doses of 125 mg and 600 mg per day, respectively and given for 21 days in 28-day cycles.

Patients underwent laboratory tests, including blood tests, and radiological tests as per routine institutional practice. This included complete blood count (CBC) and serum biochemistry prior to starting CDK 4/6 inhibitor, CBC prior to starting each subsequent cycle, ECG prior to starting ribociclib and then as clinically indicated. Radiological assessment was done with either CT scan or PET-CT scan prior to starting CDK 4/6 inhibitor and once every 3–6 cycles thereafter.

The analysis was restricted to patients who started the CDK 4/6 inhibitor between January 2016 and June 2019. Patients' records, including electronic medical records of Tata Memorial Centre, Mumbai, were used to extract data according to a pre-defined case record form. All data were not fully anonymized when we accessed the records. Extracted data included the demographic information, clinical examination findings, Eastern Cooperative Oncology Group Performance Status (ECOG PS), treatment details prior to starting CDK 4/6 inhibitor therapy, partner endocrine treatment, dose modifications of CDK 4/6 inhibitor if any, radiological reports, disease status at various time points after starting CDK 4/6 inhibitor, toxicity, and death.

The efficacy endpoints were progression-free (PFS) and overall survival (OS). PFS was defined as the time interval in months between the date of starting CDK 4/6 inhibitor and date of first documented clinical and/or radiological disease progression or death due to any cause, whichever was earlier. OS was defined as the time interval in months between the date of starting CDK 4/6 inhibitor and death due to any cause. Patients who did not experience the events for PFS and OS on the data cutoff date were censored. The date of first progression was extracted from clinical and radiological records and mostly followed RECIST version 1.1 criteria but was not reconfirmed as part of this analysis. Adverse events were extracted from medical records and laboratory reports for the period that patients were on CDK 4/6 inhibitor treatment and classified according to CTCAE version 4.03 [17].

## Statistical analysis

Demographic variables and toxicities were descriptively reported using median and interquartile range for continuous variables and frequency and proportion for categorical data. Survival outcomes (PFS and OS) were analysed by the Kaplan-Meier method and log rank test was used to compare outcomes between groups. Multivariable Cox regression analysis was performed with inclusion of covariates that significantly impacted outcome in univariable analysis. Data was analysed using IBM SPSS Statistics version 25 and R Studio version 1.2.5019.

## Results

### Patient characteristics

A total of 101 patients with median age of 57 (IQR, 48–62) years were included in the study, of whom 22 (21.8%) and 79 (78.2%) received CDK 4/6 inhibitor in first line and second- or later-line settings, respectively. The characteristics of the included patients are shown in Table 1. Of note, 80 (79.2%) patients were postmenopausal, 15 (14.9%) patients were in ECOG PS 3, 95 (94.1%) patients had invasive ductal carcinoma, 6 (5.9%) had invasive lobular carcinoma, 88 (87.1%) had visceral metastases, 13 (12.9%) had bone-only metastasis and the median hemoglobin was 10.8 (IQR, 10.0–11.5) g/dl. On the data cutoff date of June 30, 2020, the median follow-up of all, first line and second or later line patients were 18.5 (0.5–43.7) months, 21.7 (0.5–41.9) months, and 17.2 (0.5–43.7) months, respectively.

### Prior treatment

This study cohort comprised a heavily pre-treated group of patients. In terms of endocrine treatment, 23 (22.8%) patients had received 1 line, 36 (35.6%) patients had received 2 lines, and 18 (17.8%) patients had received 3 lines of endocrine therapy prior to receiving CDK 4/6 inhibitors. In terms of chemotherapy, 21 (20.8%) patients had received 1 line, 20 (19.8%) patients had received 2 lines, 23 (22.8%) patients had received 3 lines, and 11 (10.9%) patients

**Table 1. Patient characteristics.**

| Baseline Clinical Characteristics | Value (n = 101) |
|---|---|
| **Age** | |
| <45 | 12 |
| 45–64 | 68 |
| 65–75 | 16 |
| >75 | 05 |
| **Median age (IQR)** | 57 (48–62) |
| **Median Hb (IQR)** | 10.8 (10.0–11.5) |
| **Median WBC (IQR)** | 5.18 (3.75–7.13) |
| **Median Platelet (IQR)** | 224 (167.5–304.0) |
| **Menopausal status** | |
| Premenopausal | 21 (20.8%) |
| Postmenopausal | 80 (79.2%) |
| **ECOG PS** | |
| 1 | 44 (43.6%) |
| 2 | 42 (41.6%) |
| > = 3 | 15 (14.9%) |
| **Histology** | |
| Invasive ductal | 95 (94.1%) |
| Invasive lobular | 06 (5.9%) |
| **ER and PR status** | |
| ER and PR positive | 85 (84.1%) |
| ER positive & PR negative | 16 (15.8%) |
| ER negative & PR positive | 00 (0.0%) |
| **ER Allred score** | |
| 3–6 | 09 (8.9%) |
| 7–8 | 92 (91.1%) |
| **PR Allred score** | |
| 3–6 | 67 (66.3%) |
| 7–8 | 34 (33.7%) |
| **Metastatic sites** | |
| Visceral metastasis | 88 (87.1%) |
| Bone only metastasis | 13 (12.9%) |
| **Line of starting CD K 4/6 inhibitors** | |
| First line | 22 (21.8%) |
| Second line | 25 (24.8%) |
| Third line | 18 (17.8%) |
| Fourth line | 14 (13.9%) |
| Fifth line | 15 (14.9%) |
| Sixth line | 07 (6.9%) |
| **Number of patients N = 101** | |
| Palbociclib | 91 (90.1%) |
| Ribociclib | 10 (9.9%) |
| **Partner drugs received N = 101** | |
| Fulvestrant | 56 (55.4%) |
| Letrozole | 26 (25.7%) |
| Exemestane | 08 (7.9%) |
| Letrozole+ leuprolide | 08 (7.9%) |
| Fulvestrant + leuprolide | 03 (2.97%) |

Abbreviations: ER, estrogen receptor; PR, progesterone receptor; PS, performance status.

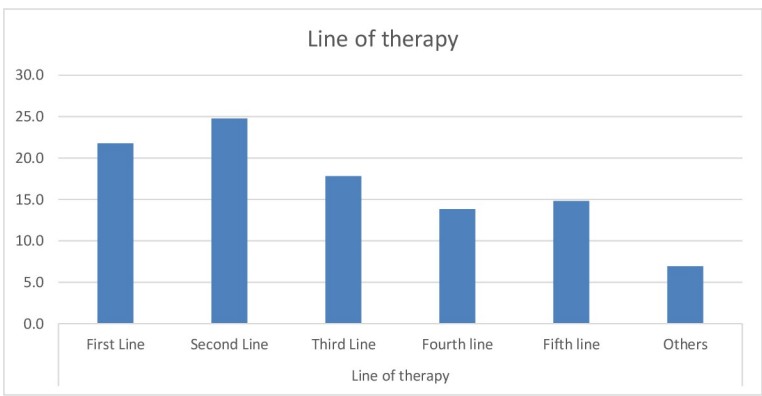

**Fig 1. Line of therapy in which CDK 4/6 inhibitor was started.**

had received 4 lines prior to starting CDK 4/6 inhibitor therapy, counting (neo) adjuvant chemotherapy as one line of treatment.

## Treatment exposure to CDK 4/6 inhibitors

Palbociclib was used in 91 (90.1%) patients while ribociclib was used in 10 (9.9%) patients. CDK 4/6 inhibitors were used in the first line setting in 22 (21.8%) patients and as second- or later-line treatment in 79 (78.2%) patients (Fig 1). The various partner drugs with palbociclib/ribociclib included fulvestrant in 59 (58.4%) patients and an aromatase inhibitor in 41 (40.6%) patients. The median number of cycles of CDK 4/6 inhibitor was 6 (range 0.5–43). CDK 4/6 inhibitor was stopped because of disease progression in 77 (76.2%) patients and toxicity or poor tolerance in 3 (2.97%) patients.

## Toxicity and dose modification

The details of hematological and non-hematological toxicities are shown in Table 2. The CDK 4/6 inhibitors were generally well tolerated. Febrile neutropenia was seen in 9 (9%) patients and grade 3–4 neutropenia in 45 (45%) patients. 75% of patients who experienced grade ¾ neutropenia had bone metastasis while 74% of patients who did not develop grade ¾ neutropenia had bone metastasis. Thus, the presence of bone metastases seems to be not correlated with the occurrence of neutropenia. Dose reduction due to toxicity was required in 32 (31.7%) patients (Table 3).

## Subsequent treatment after progression on CDK 4/6 inhibitors

The most frequent treatment after CDK 4/6 inhibitor was chemotherapy in 30 (29.7%) patients, followed by combination of chemotherapy and endocrine therapy in 19 (18.8%) patients and endocrine therapy in 10 (9.9%) patients. Eighteen (17.8%) patients could not receive any further cancer directed treatment and were planned for only symptomatic and palliative care (Table 4).

## Response

The best response to CDK 4/6 inhibitor treatment was complete response in 1 (1.0%) patient, partial response in 52 (51.5%) patients, stable disease in 19 (18.8%) patients, disease progression in 28 (27.7%) patients, and unavailable in 1 (1.0%) patient.

**Table 2. Adverse events due to palbociclib and ribociclib.**

| Event | Palbociclib N = 90 | | Ribociclib N = 10 | | Total patients N = 100 | |
|---|---|---|---|---|---|---|
| | Any grade Number of patients (percent) | Grade 3/ 4 | Any grade Number of patients (percent) | Grade 3 /4 | Any grade Number of patients (percent) | Grade 3 / 4 |
| Anaemia | 90 (100) | 29 (32) | 10 (100) | 1 (10) | 100 (100) | 30 (30) |
| Thrombocytopenia | 73 (81) | 19 (21) | 10 (100) | 0 | 83 (83) | 19 (19) |
| Neutropenia | 81 (90) | 42 (47) | 10 (100) | 3 (30) | 91 (91) | 45 (45) |
| Febrile neutropenia | 9 (10) | 9 (10) | 0 | 0 | 9 (9) | 9 (9) |
| Diarrhoea | 63 (70) | 6 (7) | 9(90) | 0 | 72 (72) | 6 (6) |
| Mucositis | 49 (54) | 2 (2) | 8 (80) | 0 | 57 (57) | 2 (2) |
| Transaminitis | 64 (71) | 4 (4) | 9 (90) | 0 | 73 (73) | 4 (4) |
| QTc prolongation | 0 | 0 | 0 | 0 | 0 | 0 |

## Survival

There were 80 PFS events and 38 deaths in the study population on the data cutoff date. The median PFS in all included patients, those who received CDK 4/6 inhibitor in first line setting, and in those who received it in second- or later-line setting were 7.7 (95% CI 5.52–11.9) months, 21.1 (95% CI 16.3-not estimable) months, and 5.98 (95% CI 4.96–7.89) months, respectively (Fig 2A and 2B). The median OS in all included patients, those who received CDK 4/6 inhibitor in first line setting, and those who received it in second- or later-line setting were 27.1 (95% CI 21.9-not estimable) months, not achieved, and 20.2 (95% CI 14.1-not estimable) months, respectively (Fig 3A and 3B). In the overall study population, the 12-month, 24-month, and 36-month OS were 73.8% (95% CI 65.0%-83.6%), 55.2% (95% CI 44.2%-68.9%), and 42.7% (95% CI 29.6%-61.6%), respectively. We performed a subgroup analysis based on endocrine sensitivity. There was no difference in median PFS between endocrine sensitive and resistant patients [7.9 months (95% CI 4.9–10.9) versus 8.2 months (95% CI 0.9–15.5), respectively].

## Univariable analysis and multivariable analysis for PFS and OS

The univariable Cox regression analysis for PFS and OS is shown in S1 Table. Multivariable Cox regression analysis for PFS and OS is shown in Table 5 and Table 6, respectively. In the PFS analysis, first line setting (HR 0.49, 95% CI 0.25–0.97, p = 0.043) and ECOG performance status 1 (HR 0.43, 95% CI 0.20–0.91, p = 0.028) were significantly associated with better outcome while in the OS analysis only ECOG PS 1 was significantly associated (HR 0.04, 95% CI 0.008–0.206, p = 0.000), with better outcome.

## Discussion

Our analysis suggests that, after accounting for differences in patient characteristics, the efficacy and toxicity results of CDK 4/6 inhibitor treatment in routine clinical practice are similar

**Table 3. Dose reductions of palbociclib and ribociclib.**

| Dose reduction level | Yes (n = 32) |
|---|---|
| 125mg to 100mg (palbociclib) | 24 (75.0%) |
| 100 mg to 75 mg(palbociclib) | 04(12.5%) |
| 125 mg to75 mg(palbociclib) | 02(6.3%) |
| 400 mg to 200 mg(Ribociclib) | 01(3.1%) |
| 600 mg to 400 mg (Ribociclib) | 01(3.1%) |

**Table 4. Subsequent treatments after progression on CDK 4/6 inhibitors.**

|  | N = 77 |
| --- | --- |
| Chemotherapy | 30 (29.7%) |
| Endocrine Therapy | 10 (9.9%) |
| Best supportive care | 18 (17.8%) |
| Chemotherapy+ endocrine therapy | 19 (18.8%) |

to those reported in pivotal clinical trials. Our results are important because the dataset includes patients who are often excluded from clinical trials, such as those with poor performance status, heavily pretreated status and extensive visceral disease. The median PFS and OS in our patient population in first line (21.1 months and not reached, respectively) and second- or later-line (5.98 and 20.2 months, respectively) settings have to be considered in this context.

Our study population was comprised of heavily pretreated metastatic patients with 53% having received two or more prior lines of endocrine therapy and/or chemotherapy in contrast to PALOMA-2 trial [2] which included patients without any prior systemic therapy for metastatic disease and PALOMA-3 trial [4] wherein 46% patients had received one prior endocrine therapy while none had received prior chemotherapy in metastatic setting. Further, an overwhelming majority of our patients (87%) had visceral disease and only 13% had bone-only disease compared with 23% patients with bone-only metastases in PALOMA-2 study. Importantly, 56% of our patients were in ECOG PS 2 or 3 in contrast to none in PALOMA-3 study [4]. These considerations suggest that our study population had poorer prognostic characteristics compared to those in pivotal trials.

As expected, our analysis suggests that patients who had better performance status and were less heavily pre-treated had better outcomes compared with those with poorer performance status and more heavy prior treatment. These findings likely reflect the impact of host characteristics and tumor resistance to multiple drugs on the outcomes, rather than the effect of initiating CDK 4/6 inhibitor at differing time points in the natural history of the disease. A similar real-world experience from Sweden also reported that the number of prior lines of chemotherapy was a significant adverse factor associated with shorter PFS [12].

The safety profile of CDK4/6 inhibitors in our patient population was concordant with that reported from pivotal clinical trials. Palbociclib was well tolerated at the 125 mg dose, with only 32% patients requiring dose reduction compared with 36% requiring dose reduction in PALOMA-2 [2] and 34% in PALOMA-3 trial [4]. The proportion of patients with grade 3 or 4 neutropenia with palbociclib was 45%, which is less compared to that reported from PALOMA-1 (62%) [1], PALOMA-2 (66%) [2, 3] and PALOMA-3 (65%) [4], studies, respectively. This discrepancy could be due to ascertainment bias in real-world data such as ours. Febrile neutropenia requiring antibiotics developed in a small minority of patients (9%) attesting to the safety of these drugs in routine outpatient clinical practice. Toxicity or intolerance requiring treatment discontinuation was seen in a small proportion of patients in our study (3.0%), comparable to that seen in PALOMA-3 (4%) [4] & MONALEESA-7 (4%), studies [7], and that reported in real-world setting from India (2.7%) [16].

Our study is unique in some aspects and different from previously published real-world data. Our patient population was heavily treated with 80% of patients having received CDK 4/6 inhibitors in 2nd or later lines, and 50% patients in 3rd or later line of treatment, had poorer prognostic characteristics with 87% with visceral disease and 56% in ECOG performance status 2–3. This is a different patient profile than that included in the pivotal trials and yet treatment was reasonably well tolerated with only 3% patients discontinuing CDK 4/6 inhibitors due to toxicity.

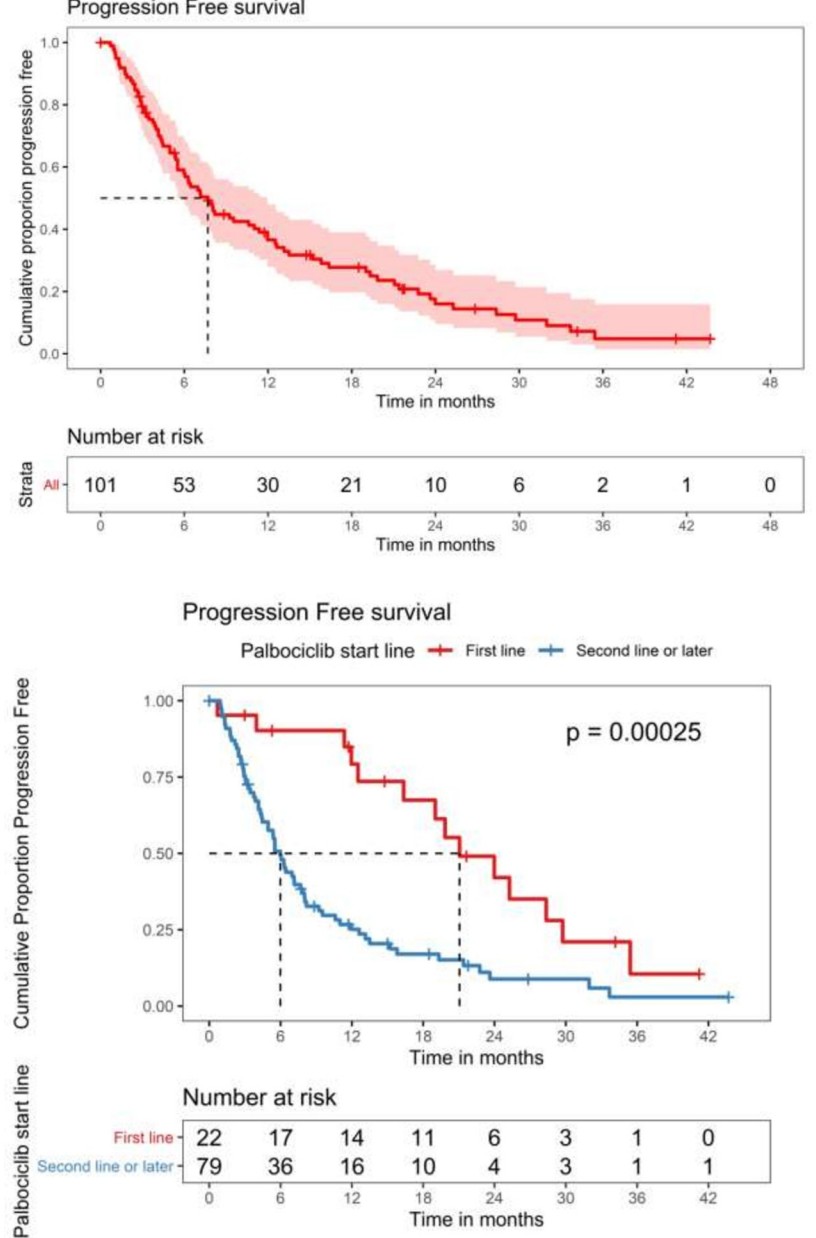

**Fig 2.** A. Progression-free survival in the overall population. B. Progression-free survival in patients who received CDK 4/6 inhibitor in first-line vs. those who received it in second or later line treatment.

Only a small fraction (20%) of our patients received CDK4/6 inhibitors as first line treatment although various guidelines recommend their use as first line therapy in hormone receptor positive metastatic breast cancer. It is worth noting that although CDK4/6 inhibitors have been approved in first line setting based on improvement in PFS, OS benefit has been unequivocally proven only in second line scenario. Therefore, use of CDK 4/6 inhibitors in second or later lines is associated with shorter duration of their use and lower cost compared to their use in first line. Most patients in our country are not covered by health insurance and have to spend out of their pocket for treatment. Treatment with CD4/6 inhibitors in 1st line setting

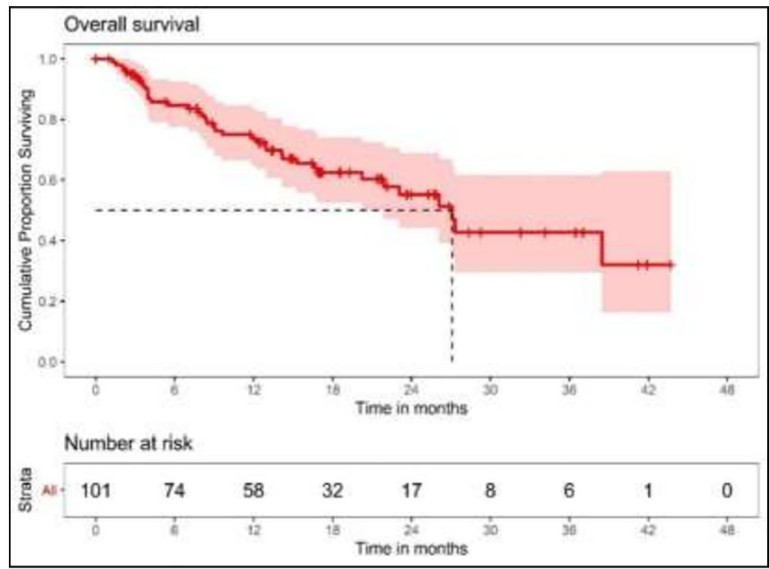

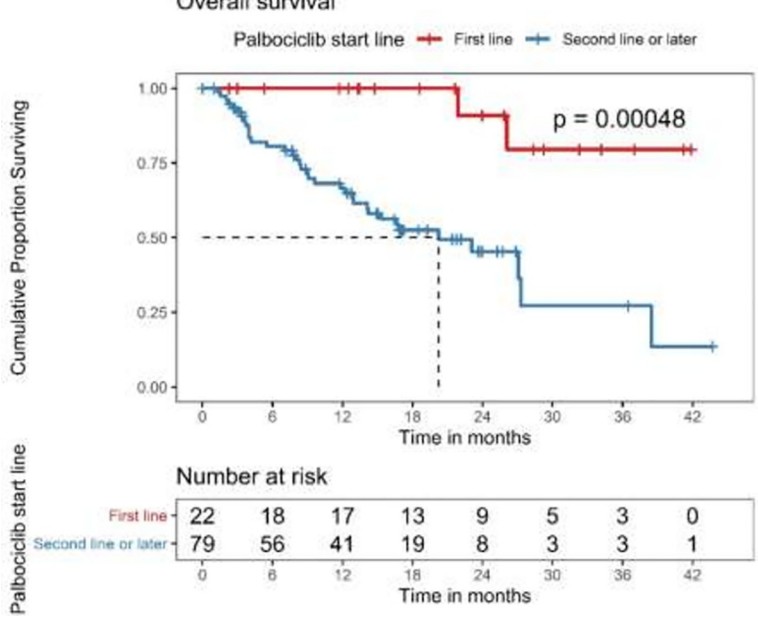

**Fig 3.** A. Overall survival in the study population. B. Overall survival in patients who received CDK 4/6 inhibitor in first-line versus those who received it in second or later line treatment.

was offered to our patients when deemed appropriate depending upon the disease burden or financial status.

Our study has some limitations including its retrospective nature, relatively small sample size, potential biases in patient selection, heterogeneous patient population and ascertainment of toxicity. Therefore, the results should be interpreted cautiously but can be used to counsel patients about expected safety and efficacy outcomes.

**Table 5. Multivariable analysis of progression-free survival.**

| Multivariate analysis for PFS | | HR | 95.0% CI for HR | | P value |
| --- | --- | --- | --- | --- | --- |
| | | | Lower | Upper | |
| ECOG PS | PS1 | 0.437 | 0.209 | 0.913 | 0.028 |
| | PS2 | 0.853 | 0.435 | 1.676 | 0.645 |
| | PS > = 3 | | | | |
| Line of therapy | First Line (Ref.) | 0.497 | 0.252 | 0.978 | 0.043 |
| | Second Line or later | | | | |
| Bone vs. Visceral | Bone only | 0.545 | 0.263 | 1.128 | 0.102 |
| | Visceral (Ref.) | | | | |
| Prior CT | No | 0.631 | 0.333 | 1.194 | 0.157 |
| | Yes (Ref.) | | | | |

Abbreviations: PS, performance status; CT, chemotherapy; PFS, progression free survival.

**Table 6. Multivariable analysis of overall survival.**

| Multivariate analysis for OS | | HR | 95.0% CI for HR | | P value |
| --- | --- | --- | --- | --- | --- |
| | | | Lower | Upper | |
| ECOG PS | PS1 | 0.041 | 0.008 | 0.206 | 0.000 |
| | PS2 | 0.551 | 0.259 | 1.173 | 0.122 |
| | PS > = 3 (Ref.) | | | | |
| Line of therapy | First Line | 0.515 | 0.095 | 2.789 | 0.441 |
| | Second Line or later (Ref.) | | | | |
| Prior CT | No | 0.382 | 0.125 | 1.170 | 0.092 |
| | Yes (Ref.) | | | | |

Abbreviations: PS, performance status; CT, chemotherapy; OS, overall survival.

## Conclusion

Palbociclib and ribociclib are well tolerated when used in routine clinical practice in Indian patient population and result in survival outcomes in first or subsequent lines of treatment that are concordant with those reported in pivotal clinical trials. Our results suggest that poor performance status at the time of initiating CDK4/6 inhibitor therapy is associated with worse overall survival and not the number of previous lines of treatment.

## Supporting information

**S1 Table. The univariable Cox regression analysis for PFS and OS.**
(DOCX)

## Author Contributions

**Conceptualization:** Sushmita Rath, Prahalad Elamarthi, Sudeep Gupta.

**Data curation:** Sushmita Rath, Prahalad Elamarthi, Seema Gulia, Ravindra Nandhana, Jyoti Bajpai, Sudeep Gupta.

**Formal analysis:** Sushmita Rath, Prahalad Elamarthi, Pallavi Parab, Seema Gulia, Ravindra Nandhana, Smruti Mokal, Sudeep Gupta.

**Methodology:** Sushmita Rath, Prahalad Elamarthi, Sudeep Gupta.

**Project administration:** Sushmita Rath, Prahalad Elamarthi, Pallavi Parab, Yogesh Kembhavi, Prema Perumal.

**Supervision:** Sushmita Rath, Seema Gulia, Jyoti Bajpai, Jaya Ghosh, Sudeep Gupta.

**Writing – original draft:** Sushmita Rath, Prahalad Elamarthi, Seema Gulia, Sudeep Gupta.

**Writing – review & editing:** Sushmita Rath, Prahalad Elamarthi, Pallavi Parab, Seema Gulia, Ravindra Nandhana, Smruti Mokal, Yogesh Kembhavi, Prema Perumal, Jyoti Bajpai, Jaya Ghosh, Sudeep Gupta.

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
