## [Decision Letter · Decision Letter 0]

19 Apr 2021

PONE-D-21-03932

Efficacy and safety of palbociclib and ribociclib in patients with estrogen and/or progesterone receptor positive, HER2 receptor negative metastatic breast cancer in routine clinical practice

PLOS ONE

Dear Dr. Gupta,

Thank you for submitting your manuscript to PLOS ONE. After careful consideration, we feel that it has merit but does not fully meet PLOS ONE’s publication criteria as it currently stands. Therefore, we invite you to submit a revised version of the manuscript that addresses the points raised during the review process.

We look forward to receiving your revised manuscript.

Kind regards,

Albiruni Abdul Razak, MB MRCPI

Academic Editor

PLOS ONE

Additional Editor Comments:

Dear Author,

After careful consideration, we would recommend the paper be revised as suggested by the 2 reviewers.

Journal Requirements:

2. In the ethics statement in the manuscript and in the online submission form, please provide additional information about the patient records/samples used in your retrospective study, including: a) whether all data were fully anonymized before you accessed them; b) the date range (month and year) during which patients' medical records/samples were accessed; c) the source of the medical records/samples analyzed in this work (e.g. hospital, institution or medical center name).

[Dr Sudeep Gupta

I have read the journal's policy and the authors of this manuscript have the following competing interests:

Institutional financial interests for conducted research: Roche, Sanofi, Johnson & Johnson, Amgen, Celltrion, Oncosten, Novartis, Intas, Eisai, Biocon, Astrazeneca

Non-remunerated activities – Advisory board: Roche, Sanofi, Dr. Reddy’s Laboratories, Biocon, Pfizer, Oncosten, Core Diagnostics, Astrazeneca

All other authors have declared no competing interests.]

Reviewers' comments:

Reviewer's Responses to Questions

**Comments to the Author**

1. Is the manuscript technically sound, and do the data support the conclusions?

Reviewer #1: Partly

Reviewer #2: Yes

2. Has the statistical analysis been performed appropriately and rigorously? 

Reviewer #1: I Don't Know

Reviewer #2: I Don't Know

3. Have the authors made all data underlying the findings in their manuscript fully available?

Reviewer #1: No

Reviewer #2: Yes

4. Is the manuscript presented in an intelligible fashion and written in standard English?

Reviewer #1: Yes

Reviewer #2: Yes

5. Review Comments to the Author

Reviewer #1: Interesting report.

I have several technical comments:

1. There are 16 actual references, but in the text there are 17 .

Due to the same miscalculation I would recommend to change efference numbers on page 6 , line 99 to 11-15, instead of 11-14.

Line 100 at the same page - 14-15, instead of 15-16 .

Line 143 on page 9 - change reference 17 to 16 .

Line 295 on page 21 - change 16 to 15

2. Table 2, on page 14 : Febrile neutropenia -there are no Grade 1 or 2 in this adverse event . Please reevaluate the numbers.

3. Table 4 on page 16 :

The total number of patients receiving subsequent treatment is 77 , and not 101.

4.Line 220 on page 16 :

You may consider to add "disease" when describing "Stable disease ".

5. Don't feel that the study provide enough information about Ribociclib safety, while only 10 patients reported.

I don`t think that there is significant value for this small sample.

If one decide to proceed with the same group of patients, I would recommend to address drug specific side effects in more detailed way. For example neutropenic fever and liver enzymes elevation for each on of the drugs, or ECG changes for Ribociclib population.

Reviewer #2: This is a retrospective review of 101 patients treated with CDK4/6 inhibitors with a focus on safety and efficacy. There has been several similar real world reports on this, however the authors do not explicitly state what is novel about this study. While there may be scant data on this in India, 80% of these patients received this treatment in 2nd line setting or later, and 50% third line or later. 87% had visceral disease and 56% had ECOG 2-3. And almost 20% were offered best supportive care after progression on this treatment. This is a heavily treated population with poorer prognostic characteristics, a very different patient profile than that seen in the PALOMA trials. And yet treatment was reasonably well tolerated with only 3% discontinuing due to toxicity which is important. While this aspect is mentioned in the author summary, it should be the focus of the paper, as it was not apparently clear from the abstract and even the title. There are much fewer studies providing real world data on CDK4/6 inhibitors in later lines as seen here.

Due to the differences between patient populations in this study and that seen in the PALOMA trials, there should be less of an emphasis on efficacy comparisons. It would be good to provide clarification on why this treatment was not offered in the 1st line / second line (was it approved at that point or was access on a compassionate basis?).

45% of patients experienced Grade 3/4 neutropenia, it would be interesting to conduct a subgroup analysis and report what percentage of these patients had bone metastases. A similar subgroup analysis reporting duration of PFS intervals in responders to previous endocrine therapy would be interesting, if there was any difference vs non responders.

Overall the study was well conducted however needs to be clear in stating how this work is different to previously reported studies evaluating safety and efficacy of CDK4/6 inhibitors in metastatic hormone receptor positive breast cancer patients.

6. PLOS authors have the option to publish the peer review history of their article (what does this mean?). If published, this will include your full peer review and any attached files.

Reviewer #1: No

Reviewer #2: No

---

## [Author Response · Author response to Decision Letter 0]

29 May 2021

Reply to Editor’s and Reviewers’ comments

We thank the Editor and reviewers for their painstaking review of our manuscript and their helpful suggestions. The incorporation of these suggestions has improved the quality of our manuscript. We hope that the revised version will be found suitable for publication.

EDITOR'S SPECIFIC COMMENTS:

Reply: We rechecked our manuscript formatting and it now meets PLOS ONE's style requirements.

2. In the ethics statement in the manuscript and in the online submission form, please provide additional information about the patient records/samples used in your retrospective study, including: a) whether all data were fully anonymized before you accessed them; b) the date range (month and year) during which patients' medical records/samples were accessed; c) the source of the medical records/samples analyzed in this work (e.g. hospital, institution or medical center name).

Reply: a) All data were not fully anonymized when we accessed the records.

b)The patients’ medical records were accessed between January 2016 and June 2019. This was mentioned in the ‘Methods’ section.

c)The medical records were analysed from the electronic medical records of Tata Memorial Centre, Mumbai.

Reply: The Competing Interest statement of the authors have been added to the cover letter

4. We note that you have indicated that data from this study are available upon request. PLOS only allows data to be available upon request if there are legal or ethical restrictions on sharing data publicly.

b) If there are no restrictions, please upload the minimal anonymized data set necessary to replicate your study findings as either Supporting Information files or to a stable, public repository and provide us with the relevant URLs, DOIs, or accession numbers

Reply: There are federal Indian regulations on providing access to data for individuals outside the country. All such requests have to be cleared by the Indian Health Ministry Screening Committee. The de identified data set will be provided to any interested researcher outside India after obtaining the required approval from the Health Ministry Screening Committee. For researchers inside India the de identified data can be provided with hospital Ethics Committee approval. 

Reviewer 1

1. There are 16 actual references, but in the text there are 17 .

Due to the same miscalculation I would recommend to change efference numbers on page 6 , line 99 to 11-15, instead of 11-14.

Line 100 at the same page - 14-15, instead of 15-16 .

Line 143 on page 9 - change reference 17 to 16 .

Line 295 on page 21 - change 16 to 15

Reply: We apologize for this error. There are 17 actual references. Reference no.11 and 12 were clubbed together as 11 inadvertently. Therefore, none of the references require a change in number.

2. Table 2, on page 14 : Febrile neutropenia -there are no Grade 1 or 2 in this adverse event . Please re-evaluate the numbers.

Reply: Agreed, we have recalculated the numbers and revised Table 2 accordingly.

3. The total number of patients receiving subsequent treatment is 77 , and not 101.

Reply : Agreed, we have changed the total number of patients receiving subsequent treatment in Table 4.

4. You may consider to add "disease" when describing "Stable disease ".

Reply : Agreed, we have added "disease" while describing Stable disease " in line 232 on page no 18.

5. Don't feel that the study provide enough information about Ribociclib safety, while only 10 patients reported.

I don`t think that there is significant value for this small sample.

If one decide to proceed with the same group of patients, I would recommend to address drug specific side effects in more detailed way. For example neutropenic fever and liver enzymes elevation for each on of the drugs, or ECG changes for Ribociclib population.

Reply: We agree with the reviewer that a definite conclusion about Ribociclib safety cannot be made from our study since only 10 patients had received this drug. It is important to note that none of the patients who received Ribociclib had ECG changes. As suggested, we have now provided adverse events to palbociclib and ribociclib separately, and accordingly revised Table 2.

Reviewer 2

1. This is a retrospective review of 101 patients treated with CDK4/6 inhibitors with a focus on safety and efficacy. There has been several similar real world reports on this, however the authors do not explicitly state what is novel about this study. While there may be scant data on this in India, 80% of these patients received this treatment in 2nd line setting or later, and 50% third line or later. 87% had visceral disease and 56% had ECOG 2-3. And almost 20% were offered best supportive care after progression on this treatment. This is a heavily treated population with poorer prognostic characteristics, a very different patient profile than that seen in the PALOMA trials. And yet treatment was reasonably well tolerated with only 3% discontinuing due to toxicity which is important. While this aspect is mentioned in the author summary, it should be the focus of the paper, as it was not apparently clear from the abstract and even the title. There are much fewer studies providing real world data on CDK4/6 inhibitors in later lines as seen here.

Reply : We thank the reviewer for the pertinent comment. We have now elaborated on this unique aspect of our cohort in the ‘Discussion’ (page no. 24, para 5, lines 320-327)

2. Due to the differences between patient populations in this study and that seen in the PALOMA trials, there should be less of an emphasis on efficacy comparisons. It would be good to provide clarification on why this treatment was not offered in the 1st line / second line (was it approved at that point or was access on a compassionate basis?).

Reply: Only a small fraction (20 %) of our patients received CDK4/6 inhibitors as first line treatment although various guidelines recommend their use as first line therapy in hormone receptor positive metastatic breast cancer. It is worth noting that although CDK4/6 inhibitors have been approved in first line setting based on improvement in PFS, OS benefit has been unequivocally proven only in second line scenario. Therefore, use of CDK4/6 inhibitors in second or later lines is associated with shorter duration of their use and lower cost compared to their use in first line. Most patients in our country are not covered by health insurance and have to spend out of their pocket for treatment. Treatment with CD4/6 inhibitors in 1st line setting was offered to our patients when deemed appropriate depending upon the disease burden or financial status. This has now been added to the ‘Discussion’ (page no. 24, para 6, lines 328-340)

3. 45% of patients experienced Grade 3/4 neutropenia, it would be interesting to conduct a subgroup analysis and report what percentage of these patients had bone metastases. A similar subgroup analysis reporting duration of PFS intervals in responders to previous endocrine therapy would be interesting, if there was any difference vs non responders.

Reply: We thank the Reviewer for this suggestion. 75% of patients who experienced grade ¾ neutropenia had bone metastasis while 74% of patients who did not develop grade ¾ neutropenia had bone metastasis. Thus the presence of bone metastases seems to be not correlated with the occurrence of neutropenia. This has now been added to the Results (page 15, para 1, lines 199- 202). 

We performed a subgroup analysis based on endocrine sensitivity. There was no difference in median PFS between endocrine sensitive and resistant patients [7.9 months (95% CI 4.9-10.9) versus 8.2 months (95% CI 0.9-15.5), respectively]. These points have now been added to the ‘Results’ (page 19, para 1, lines 248-251).

4. Overall the study was well conducted however needs to be clear in stating how this work is different to previously reported studies evaluating safety and efficacy of CDK4/6 inhibitors in metastatic hormone receptor positive breast cancer patients.

Reply: We thank the reviewer for the encouraging comment. Our study population is more heavily pre-treated and with poorer performance status compared to earlier studies. We have shown that poorer performance status, at the beginning of CDK4/6 inhibitor therapy, is associated with worse overall survival and not the line of therapy.

---

## [Editor Report · Decision Letter 1]

11 Jun 2021

Efficacy and safety of palbociclib and ribociclib in patients with estrogen and/or progesterone receptor positive, HER2 receptor negative metastatic breast cancer in routine clinical practice

PONE-D-21-03932R1

Dear Dr. Gupta,

We’re pleased to inform you that your manuscript has been judged scientifically suitable for publication and will be formally accepted for publication once it meets all outstanding technical requirements.

Kind regards,

Albiruni Abdul Razak, MB MRCPI

Academic Editor

PLOS ONE

Additional Editor Comments (optional):

Dear Authors,

You have responded satisfactorily to the suggestions and comments by the reviewers.

No further suggestions from me

The only outstanding issue, to my mind, is the response to the Editor re: the need for Indian Ministry of Health approval for data release if the need ever arises.

Academic Editor PLOS ONE.
---

## [Editor Report · Acceptance letter]

14 Jul 2021

PONE-D-21-03932R1 

Efficacy and safety of palbociclib and ribociclib in patients with estrogen and/or progesterone receptor positive, HER2 receptor negative metastatic breast cancer in routine clinical practice 

Dear Dr. Gupta:

I'm pleased to inform you that your manuscript has been deemed suitable for publication in PLOS ONE. Congratulations! Your manuscript is now with our production department. 

Kind regards, 

on behalf of

Dr. Albiruni Abdul Razak 

Academic Editor

PLOS ONE